# Generation and Characterization of Stable Small Colony Variants of USA300 *Staphylococcus aureus* in RAW 264.7 Murine Macrophages

**DOI:** 10.3390/antibiotics13030264

**Published:** 2024-03-16

**Authors:** Dalida Bivona, Carmelo Bonomo, Lorenzo Colombini, Paolo G. Bonacci, Grete F. Privitera, Giuseppe Caruso, Filippo Caraci, Francesco Santoro, Nicolò Musso, Dafne Bongiorno, Francesco Iannelli, Stefania Stefani

**Affiliations:** 1Medical Molecular Microbiology and Antibiotic Resistance Laboratory (MMARLab), Department of Biomedical and Biotechnological Sciences (BIOMETEC), University of Catania, 95123 Catania, Italy; dalida.bivona@phd.unict.it (D.B.); carmelo.bonomo@phd.unict.it (C.B.); paolo.bonacci@phd.unict.it (P.G.B.); stefania.stefani@unict.it (S.S.); 2Laboratory of Molecular Microbiology and Biotechnology, Department of Medical Biotechnologies, University of Siena, 53100 Siena, Italy; lorenzo.colombini2@unisi.it (L.C.); francesco.iannelli@unisi.it (F.I.); 3Department of Clinical and Experimental Medicine, University of Catania, 95123 Catania, Italy; greteprivitera@gmail.com; 4Department of Drug and Health Sciences, University of Catania, 95125 Catania, Italy; giuseppe.caruso2@unict.it (G.C.); carafil@unict.it (F.C.); 5Oasi Research Institute-IRCCS, 94018 Troina, Italy; 6Biochemical Section, Department of Biomedical and Biotechnological Sciences (BIOMETEC), University of Catania, 95123 Catania, Italy

**Keywords:** persistence, carnosine, erythromycin, macrophages, USA300, methicillin-resistant *S. aureus*

## Abstract

Intracellular survival and immune evasion are typical features of staphylococcal infections. USA300 is a major clone of methicillin-resistant *S. aureus* (MRSA), a community- and hospital-acquired pathogen capable of disseminating throughout the body and evading the immune system. Carnosine is an endogenous dipeptide characterized by antioxidant and anti-inflammatory properties acting on the peripheral (macrophages) and tissue-resident (microglia) immune system. In this work, RAW 264.7 murine macrophages were infected with the USA300 ATCC BAA-1556 *S. aureus* strain and treated with 20 mM carnosine and/or 32 mg/L erythromycin. Stable small colony variant (SCV) formation on blood agar medium was obtained after 48 h of combined treatment. Whole genome sequencing of the BAA-1556 strain and its stable derivative SCVs when combining Illumina and nanopore technologies revealed three single nucleotide differences, including a nonsense mutation in the shikimate kinase gene *aroK*. Gene expression analysis showed a significant up-regulation of the *uhpt* and *sdrE* genes in the stable SCVs compared with the wild-type, likely involved in adaptation to the intracellular *milieu*.

## 1. Introduction

*Staphylococcus aureus* is able to modulate a wide set of virulence factors and resistance genes, making it an opportunistic pathogen [1]. Emerging in the past as a nosocomial pathogen, methicillin-resistant *S. aureus* (MRSA) has quickly spread in the community, following different evolutive paths through the sequential acquisition of mobile genetic elements (MGEs) [2].

A major clinical issue related to *S. aureus* is represented by persistent infections (e.g., prosthetic joint infections) characterized by an asymptomatic phase with relapses occurring months to years after the apparent resolution of the infection [3,4].

*S. aureus* has shown the ability to internalize inside non-phagocytic cells, i.e., osteoblasts and epithelial and endothelial cells and resist the phagocytosis exerted by neutrophils and macrophages, performing phagosomal escape [5,6]. 

It has also been demonstrated that *S. aureus* belonging to different genetic backgrounds is able to differently modify the expression of host genes involved in inflammation, cell metabolism, and oxidative stress [7], with all of these mechanisms confirming its persistent behavior. Intracellular survival may lead to the development of *S. aureus* reservoirs in humans, contributing to both persistence and dissemination from the infection site to other body areas [8,9,10].

In addition, these kind of infections have often been associated with a particular *S. aureus* phenotype called small colony variant (SCV), an alternative bacterial lifestyle generally related to adaptation and persistence in unfavorable environmental conditions [4]. Furthermore, the role of professional phagocytes as “trojan horses” has been largely observed in several infectious diseases, such as those caused by bacteria, fungi, and viruses. Interestingly, they are also involved in the carriage of pathogens across the blood–brain barrier [11,12]. 

Understanding the intracellular survival mechanisms of different strains of *S. aureus* is a key factor for developing effective therapeutic strategies against infections that are difficult to treat. To the best of our knowledge, the intracellular survival of *S. aureus* varies depending on the set of virulence factors produced (e.g., PVL toxin, alpha toxin, etc.) as well as its genetic background. This finding was already assessed through internalization assays in MG-63 human osteoblast cells, which revealed differences in the internalization and intracellular persistence of *S. aureus* based on its sequence type (ST) [13].

Carnosine, a dipeptide composed of β-alanine and L-histidine and found in several tissues [14,15,16], was used for its antioxidant properties and for confirming its potential role in persistence. The natural dipeptide carnosine may represent a choice in addition to the antibiotics currently used [15]. Carnosine possesses a well-known multimodal mechanism of action that includes antioxidant and anti-inflammatory properties [17,18]. It also possesses a well-documented ability to enhance both antioxidant machinery [19] and energy metabolism [20,21] in different cell types, including macrophages and microglia. Furthermore, an “anti-carnosine activity” has been found to be widely spread among bacteria, underlining that this activity could play a significant role in the phenomenon of persistence, a role that has already been proven in a rat model staphylococcal infection [22]. Lastly, carnosine analogs (e.g., homocarnosine) [23] or its chelated compounds (e.g., zinc carnosine—gastrozin) [24] have shown to act synergistically with antibiotics to counteract bacterial infections.

The present study evaluated the ability of the USA300 *S. aureus* ATCC BAA-1556 strain, maintained in our laboratory and named ATCC BAA-1556_Catania, to survive and form SCVs inside RAW 264.7 murine macrophages in the presence of the natural antioxidant carnosine and/or erythromycin. A stable SCV strain was characterized by whole genome sequencing and quantitative gene expression analysis.

## 2. Results

### 2.1. Carnosine Does Not Show Antimicrobial Activity on ATCC BAA-1556_Catania 

The antimicrobial activity of carnosine was tested at concentrations ranging from 0.15 mg/L to 160 mg/L in two media, showing a lack of inhibitory activity on *S. aureus* ATCC BAA-1556_Catania. The MIC value for erythromycin was 32 mg/L both in cation-adjusted Mueller–Hinton broth and DMEM medium, and the addition of 20 mM carnosine did not show synergic or antagonistic activity. 

### 2.2. Macrophages Infection and Evaluation of Small Colony Variants (SCVs) Formation and Stability

RAW 264.7 murine macrophages were infected with a wild-type BAA-1556_Catania *S. aureus* strain and treated with 20 mM carnosine and/or 32 mg/L erythromycin. Microphotographs of RAW 264.7 murine macrophages were acquired, depicting both those that were non-infected (Figure 1a) and infected with *S. aureus* USA300 ATCC BAA-1556_Catania (Figure 1b). As shown in Figure 1b, a considerable change in RAW 264.7 murine macrophages polarization is evident when bacteria were added.

A previous study of Caruso et al. reported that the use of carnosine on RAW264.7 murine macrophages protects the cells of macrophages, modulates the oxidative stress, and decreases the expression of genes related to inflammation and pro- and antioxidant systems [25]. According to these findings, the treatment of RAW 264.7 murine macrophages with carnosine was finalized to counteract the oxidative stress and support the macrophage counteracting the infectious process. 

At 3 h, 24 h, and 48 h, post-infection cells were lysed, and the cell lysate was plated on blood agar plates to evaluate the formation of SCVs (Figure 2). Carnosine treatment increased the formation of SCVs at each investigated time point (*p*-value = 0.019, 0.004, and <0.0001 at 3, 24, and 48 h post-infection, respectively), while treatment with erythromycin alone or in combination with carnosine significantly decreased the formation of SCVs at 24 h (*p*-value = 0.001 and <0.0001, respectively) and 48 h (*p*-value = 0.004 and <0.0001, respectively) post-infection. The comprehensive colony count is presented in Appendix A.

Stability of the isolated SCVs was evaluated using serial passages on blood agar plates. Defining the SCV phenotype as stable is disputable for the needed number of passages on BA [26,27]. Based on this assumption, the SCV phenotype of colonies isolated after treatment with the combination of carnosine and erythromycin for 48 h was maintained for several passages, whereas it was lost at the first passage in all other isolated colonies. 

The stable SCVs, isolated after 48 h post-infection from the combined treatment condition, also showed a reduced ability to cause hemolysis on blood agar medium (Figure 3) and increased MIC values for aminoglycosides. In particular, for amikacin, gentamycin, tobramycin, and kanamycin, an increase from 4 mg/L to 96 mg/L, from 0.5 mg/L to 4 mg/L, from 0.5 mg/L to 6 mg/L, and from 1 to 16 mg/L, for amikacin, gentamycin, tobramycin, and kanamycin was observed, respectively (Appendix A). 

### 2.3. Genome Analysis of SCV 

To investigate whether the SCV phenotype was associated with a specific genotype, the wild-type BAA-1556_Catania and a representative stable SCV derivative were sequenced using both Illumina (MiSeq platform, San Diego, CA, USA) and nanopore technologies (Oxford Nanopore Technologies, Oxford, UK). The sequence analysis showed that both genomes consist of a 2,878,871-bp single circular chromosome and three plasmids that were 37,136-bp, 4439-bp, and 3125-bp in length. Virulence and antibiotic resistance gene content is reported in Appendix A. The assembled genome of *S. aureus* BAA-1556_Catania showed differences at repeated regions (e.g., ribosomal RNA genes, intergenic regions) from the sequences deposited in Genbank (acc. no. CP000255-258) and on the ATCC website (https://www.atcc.org/products/baa-1556 (accessed on 7 September 2023)), possibly due to assembly errors. The genomic comparison of BAA-1556_Catania to its SCV revealed three chromosomally located nucleotide changes (Table 1). The T→A transversion (nucleotide position 532,783) in the *purR* gene caused a missense mutation (I267N) of the predicted gene product, which acts as repressor of the *pur* operon involved in purine biosynthesis [28]; the C→T transition (position 909,332) occurred in an intergenic region between RZS17_04450 and RZS17_04455; and the G→A transition in the coding sequence of *aroK* (position 1,659,182) caused a nonsense mutation, which introduced a premature stop codon. The *aroK* gene codes for the 175 amino acid shikimate kinase that is involved in the synthesis of chorismate, which is a precursor of aromatic amino acids and of the electron acceptors ubiquinone/menaquinone [29]. The mutation produced a truncated shikimate kinase (W102*). Additionally, in BAA-1556_Catania, the *hslO* gene showed a polymorphic nucleotide site (position 550,154) harboring either an A or a G without affecting the predicted gene product sequence.

### 2.4. Comparative Gene Expression Analysis

The gene expression of 10 selected genes belonging to different functional categories was analyzed by qPCR in the stable SCVs compared with the parental *S. aureus* BAA-1556_Catania. The phenotypical stability of SCVs collected 48 h post-infection from the combined treatment (erythromycin plus carnosine) allowed for their selection for expression studies instead of SCVs isolated from the other experimental conditions, owing to the transient SCV phenotype exhibited.

The investigated genes included: the *sarA*, *sigB*, and *agrA* transcriptional regulators; the *psmA*, *hla*, and *hld* virulence factors; the *sdrE* surface protein gene; and the *pdhA*, *fumC*, and *uhpt* genes involved in glucose metabolism.

Among the transcriptional regulator genes, a non-statistically significant trend of up-regulation was observed for *sarA* and *agrA* genes. A non-significant upregulation was also observed for the *hla* and *hld* genes. Finally, a significant upregulation of the *sdrE* and *uhpt* gens was detected in the SCV (*p* value < 0.05), while there was a non-significant upregulation of *fumC* (Figure 4, Appendix A). 

## 3. Discussion

Several *S. aureus* strains exploit their ability to penetrate non-phagocytic cells, inducing intracellular toxicity and tissue destruction; on the other hand, bacterial persistence in cells is believed to lead to immune and therapies failure other than chronic infections. Furthermore, the phagocytosis exerted by neutrophils and macrophages has been linked to the possible dissemination of pathogens [30], representing an additional point to be considered. 

In this paper, we investigated the mechanisms responsible for the transition of *S. aureus* ATCC BAA-1556_Catania from wild-type to SCV in RAW 264.7 murine macrophages in the presence of carnosine, erythromycin, or erythromycin plus carnosine. SCVs formation was observed in all tested conditions, thus indicating that *S. aureus* BAA-1556 has good intracellular survival and adaptation ability. Indeed, the internalization process inside RAW 264.7 murine macrophages led to SCV formation even in untreated cells, thus highlighting the tendency of *S. aureus* BAA-1556 to develop a persistent form. 

Based on this assumption, for each time point (3 h, 24 h, 48 h), the statistically significant SCV formation was considered while comparing the untreated cells with the other experimental conditions. Carnosine treatment significantly increased the formation of SCVs at each investigated time point, while treatment with erythromycin alone or in combination with carnosine significantly decreased the formation of SCVs at 24 h and 48 h post-infection. In this context, a stable SCV phenotype was observed after 48 h of exposure to the combined treatment, thus suggesting that the combination of carnosine and erythromycin triggers the mechanism underlying SCV formation, leading to longer stability. In contrast, a transient SCV phenotype was observed under the same treatment conditions (carnosine *plus* erythromycin) at other experimental time points, 3 and 24 h, respectively.

Phenotypical stability, assessed by selecting 10% of the total SCVs and performing several passages on blood agar, demonstrated the acquisition of the phenotype.

A stable SCV phenotype is usually related to mutations in specific genes involving the electron transport chain, such as *hemB*, *menD*, *ctaA*, and *thyA* [31,32]. The *menD* gene encodes 2-succinyl-6-hydroxy-2,4-cyclohexadiene-1-carboxylate synthase involved in the menadione biosynthesis pathway, located downstream the shikimate biosynthetic pathway. Our genomic analysis revealed three nucleotide changes between the wild-type *S. aureus* BAA-1556_Catania and its stable SCV derivative, of which one caused a nonsense mutation in the *aroK* gene coding for a shikimate kinase involved in the shikimate metabolic pathway and one led to a missense mutation in the *purR* gene coding for the Pur operon repressor. A nonsense point mutation in *aroD* has previously been reported in an SCV selected using subinhibitory concentrations of kanamycin. *aroD* codes for a 3-dehydroquinate dehydratase, which catalyzes the synthesis of shikimate. The shikimate pathway is essential for the subsequent synthesis of aromatic amino acids and the electron acceptors ubiquinone/menaquinone, and the *aroD* mutation was demonstrated to impair the whole pathway. It is likely that the *aroK* nonsense mutation of our SCV also impairs the shikimate pathway with similar effects. Mutations in the *S. aureus purR* gene result in the upregulation of purine biosynthetic genes but also in increased expression of fibronectin binding proteins, enhancing the pathogenic potential of *S. aureus* strains [33].

Finally, gene expression analysis was performed to elucidate the differences between the wild-type and SCV phenotype of *S. aureus* BAA-1556, focusing on target genes involved in transcriptional regulation, virulence, and metabolism. Because increased glycolytic metabolism was reported for stable SCV [34,35], genes related to glycolysis were also investigated. 

The observed over-expression of *agrA* in stable SCVs may be explained by the parallel over-expression of the *sarA* gene, and this expression profile has been previously associated with *S. aureus* intracellular survival in macrophages [36]. In turn, SarA over-expression led to a significant over-expression of *sdrE* and virulence genes (*hla* and *hld* genes), thus favoring the mechanism of adhesion (a typical strategy in the initial stages of the internalization process, when *S. aureus* shows a slowdown in metabolic activity) and persistence in macrophages [37].

As previously demonstrated, the glycolytic and fermentative pathways are up-regulated in *S. aureus* SCVs, even in the presence of oxygen, thus exhibiting a type of anaerobic metabolism [38]. Indeed, statistically significant over-expression of the *uhpt* gene encoding a hexose phosphate transporter was observed. Moreover, a trend towards over-expression was shown for the *fumC* gene encoding the fumarate dehydrogenase, responsible for the conversion of fumarate into malate. Fumarate is known to be essential for epigenetic changes associated with induced immunity, and its depletion, due to *fumC* up-regulation, is critical for the intracellular survival of SCVs [34,35].

Because chronic and persistent infections are a major clinical burden due to the failure of prolonged antibiotic treatment, our findings allow us to improve the knowledge of antibiotic resistance mechanisms, which are sometimes related to the most difficult-to-treat SCV phenotypes.

Although the intracellular survival of *S. aureus* in macrophages has already been described, there are no exhaustive studies that unambiguously describe the ability of macrophages to clear *S.aureus* and/or SCVs over time. Tuchscherr et al. [4] demonstrated the ability of the macrophages to clear *S. aureus* with no evidence of SCV formation. 

In contrast, Stoneham et al. [39] demonstrated that wild-type *S. aureus* has a survival advantage over SCVs in macrophages during prolonged infection; thus, a better clearance of SCVs was observed. 

To the best of our knowledge, there are no studies that highlight how and whether SCVs can arise inside macrophages during prolonged exposure to antibiotic anti-staphylococcal and/or antioxidant molecules. 

Genetic background can affect the ability of *S. aureus* to internalize and persist in non-phagocytic cells [7]. The strong ability of *S. aureus* BAA-1556_Catania to survive and persist inside RAW 264.7 murine macrophages could be strictly related to its genetic features; indeed, our strain belongs to the USA300 lineage, a dominant clone of community-acquired MRSA, which is extremely virulent and prone to adaptation and is involved in a wide range of human infection [40,41].

In conclusion, the overall results reported in the present study demonstrated the ability of *S. aureus* BAA-1556_Catania to form SCVs inside RAW 264.7 murine macrophages in the presence and absence of erythromycin and/or carnosine, as a stress response mechanism, and to develop a stable SCV phenotype when the combined treatment of carnosine *plus* erythromycin was extended up to 48 h. These data suggested that prolonged exposure to the combined treatment over time resulted in the acquisition of the new phenotype, facilitating the adaptability and persistence of the bacterial strain which, acquiring new mutations, demonstrated high genome plasticity.

## 4. Materials and Methods

### 4.1. Bacterial Strain

The *S. aureus* subspecies *aureus* Rosenbach BAA-1556 strain, belonging to the USA300 clone, was purchased at ATCC and used for the internalization study. The USA300 *S. aureus* clone belongs to the spa type t008 and MLST type ST8, is a Panton–Valentine Leucocidin (PVL) producer, and harbors the staphylococcal cassette chromosome *mec* (SCC*mec*) type IV and arginine catabolic mobile element (ACME) [42].

### 4.2. Determination of Minimum Inhibitory Concentration 

Minimal inhibitory concentration (MIC) was assessed by a broth microdilution method as recommended by the EUCAST v 12.0 guidelines [43], as already described [5]. Carnosine (Cat. No C9625, Sigma-Aldrich-Merck, Darmstadt, Germany) concentrations ranged from 160 mg/L to 0.15 mg/L. Erythromycin (Cat. No E5389-1G, Sigma-Aldrich-Merck, Germany) concentrations ranged from 256 mg/L to 0.25 mg/L. For the combination assay, carnosine was used at the concentration of 4.5 mg/L (20 mM), while erythromycin was used from 256 mg/L to 0.25 mg/L. MIC assays were performed in cation-adjusted Mueller–Hinton broth (CA-MHB) (Cat. No 11703503, BD Difco, Milano, Italy) and Dulbecco’s Modified Eagle Medium high glucose (DMEM high glucose) (Cat. No 11965092, Gibco), which were used to grow RAW 264.7 (ATCC TIB-71) mouse macrophage cells. Susceptibility to aminoglycosides (i.e., amikacin, gentamicin, tobramycin, and kanamycin) was assessed using E-test MIC test strips (LIOFILCHEM^®^ S.r.l., Teramo, Italy) on Mueller–Hinton Agar (Cat. No CM0337B, Oxoid, ThermoFisher Scientific, Rodano, Italy) as recommended by the EUCAST v 12.0 guidelines [43].

### 4.3. Infection Assay

#### 4.3.1. Eukaryotic Cell Culture Preparation

RAW 264.7 cells were selected to perform the infection assay. Cells were grown in 75 cm^2^ flasks in high-glucose DMEM supplemented with HEPES (Cat. No 11965092, Gibco), GlutaMAX™ Supplement, 10% Fetal Bovine Serum (FBS) (Cat. No F7524, Sigma-Aldrich-Merck), and 100 U/mL of Penicillin/Streptomycin (Cat. No 15140148, Gibco ThermoFisher Scientific). The cell culture was incubated at 37 °C and CO_2_ 5%. The medium was refreshed twice weekly. A total of 5 × 10^4^ RAW 264.7 cells were plated in a 24-well plate 24 h before infection in DMEM without penicillin/streptomycin, supplemented with 10% FBS and treated with 20 mM carnosine. 

#### 4.3.2. Infection of RAW 264.7 Cells

RAW 264.7 cells were infected with BAA-1556 *S. aureus* at a multiplicity of infection (MOI) of 50:1 (2.5 × 10^6^ bacteria) for 3 h, 24 h, and 48 h. Cells were treated with carnosine (20 mM) and/or erythromycin (32 mg/L) or left untreated as a control. 

One hour before each time point, infected cells were treated with 100 mg/mL lysostaphin (cat. No. L7386-15MG; Sigma-Aldrich, Germany) at 37 °C for 1 h to lyse bacterial cells in the extracellular environment. To confirm the absence of extracellular bacteria, 100 µL of each culture were plated on tryptic soy agar (TSA) (Cat. No 11973752, Oxoid, Thermo Scientific) plates and incubated for 16/18 h at 37 °C [5].

Osmotic lysis was performed by incubating cells in 500 µL of sterile water at 37 °C for 30 min. 

Cellular lysates, containing internalized bacteria, were diluted and plated on blood agar plates (Cat. No PB0114A, Oxoid, ThermoFisher Scientific, Italy) to count wild-type and SCVs colonies after 16/18 h of incubation at 37 °C (Figure 5). Pictures of the infected cells at the different time points were taken with a Leica DMI 4000B inverted microscope at 20× magnification.

### 4.4. Evaluation of SCV Stability

Because the coexistence of wild-type and SCV phenotypes was detected in all experimental conditions, 10% of the isolated SCVs was sub-cultured several times on blood agar plates to evaluate the phenotypic stability. SCV stability was also tested after freezing in tryptic soy broth (Cat. No CM0129R, Oxoid, ThermoFisher Scientific) containing 15% glycerol (Cat. No A16205.AP, Oxoid, ThermoFisher Scientific) and subsequent culture on blood agar plates. SCVs were defined as stable if no transition to the wild-type phenotype took place after at least six passages on blood agar plates (Figure 5). SCVs were also characterized for their ability to perform hemolysis on blood agar plates.

### 4.5. Genomic Analysis

#### 4.5.1. DNA Extraction 

*S. aureus* ATCC BAA-1556 wild-type and its stable SCVs isolated from the treatment with carnosine (20 mM) and erythromycin (32 mg/L) were selected for DNA isolation. For Illumina sequencing, genomic DNA was extracted and quantified as previously described [44]. For nanopore sequencing, high-molecular weight genomic DNA was obtained using a raffinose-based method as described [45,46].

#### 4.5.2. Genome Sequencing

Illumina sequencing was performed on the Illumina MiSeq platform (2 × 150 bp paired-end sequencing); library preparation and quantification were performed as reported, starting from 10 ng of genomic DNA [44]. The sample sheet was created using the Local Run Manager v3 software, following the instructions in the Local Run Manager v3 Software Guide provided by Illumina. 

Nanopore sequencing was carried out essentially as already described [47]. Briefly, a DNA size selection of genomic DNA was performed with 0.5 volumes of AMPure XP beads (Beckman Coulter, Milano, Italy). Following this, 2.5 μg of size-selected DNA was employed for library construction with the SQK-LSK 109 kit (Oxford Nanopore Technologies, Oxford, UK) and sequenced on a GridION X5 device (Oxford Nanopore Technologies). Real-time base calling was performed with Guppy v7.1.4 (Oxford Nanopore Technology), filtering out reads with a quality cut-off of <Q9.

#### 4.5.3. Bioinformatic Analysis

Paired-end Illumina raw reads were first trimmed with TrimGalore (v0.5.0) [48,49] to remove adapter sequences. Genome assembly was performed using both nanopore and Illumina reads as reported [50]. Assembly completeness was assessed with Bandage v.0.8.1, whereas assembly quality was evaluated with both Ideel v5.5.4 (https://github.com/mw55309/ideel (accessed on 4 September 2023)) and CheckM v1.1.3 (https://github.com/Ecogenomics/CheckM (accessed on 4 September 2023)). Bwa v0.7.17 (https://github.com/lh3/bwa (accessed on 6 September 2023)) and minimap2 v2.13 (https://github.com/lh3/minimap2 (accessed on 6 September 2023)) were used to align Illumina reads and nanopore reads to the assembled genome, respectively. Read genome mapping was visualized with Tablet v1.17.08.17 (https://github.com/cropgeeks/tablet (accessed on 6 September 2023)) and used to further verify the assembled structure. Annotation was performed with the NCBI Prokaryotic Genome Annotation Pipeline (PGAP) v5.1. Genome comparison between wild-type and SCV was performed with the following tools: (i) MUMmer v3.23 (https://github.com/mummer4/mummer (accessed on 6 September 2023)), (ii) the Artemis and Artemis Comparison Tool (ACT), and (iii) Blast (https://blast.ncbi.nlm.nih.gov/Blast.cgi (accessed on 6 September 2023)). Variant calling was performed using Medaka v0.7.1 with the option “medaka_haploid_variant” using both nanopore and Illumina reads and considering “quality score>=25”. Putative chromosomal structural variants were investigated using the Sniffles v1.0.12 structural variation caller (https://github.com/fritzsedlazeck/Sniffles (accessed on 7 September 2023)). Default parameters were used for all tools unless otherwise specified.

#### 4.5.4. Data Availability Statement

The sequencing data presented in this study can be found in the NCBI repository under the BioProject accession number PRJNA912391.

### 4.6. Gene Expression Study

#### 4.6.1. Primer Design

Primer sequences for the gene expression analysis of *pdhA* and *fumC* were designed with the primer designing tool of NCBI (https://www.ncbi.nlm.nih.gov/tools/primer-blast/; accessed on 15 June 2022) using *S. aureus* USA300_FPR3757 (GenBank accession number: NC_007793) as the reference genome. Gene expression of the regulatory genes *sarA*, *sigB*, and *agrA*; virulence genes *psmA*, *hla*, and *hld*; adhesion gene *sdrE*; and the hexose phosphate transportation gene *uhpt* was performed using previously reported primers [13].

#### 4.6.2. RNA Extraction

RNA was purified from *S. aureus* ATCC BAA-1556_Catania and its stable SCV at 48 h post-infection of RAW 264.7 cells treated with carnosine (20 mM) and erythromycin (32 mg/L); RNA quality was assessed as previously described [13].

#### 4.6.3. RT-qPCR

cDNA was synthetized with 100 ng of RNA as a template, as previously described, and RNA normalization was performed in order to obtain a final cDNA concentration equal to 8 ng/µL using the QuantiNova SYBR Green PCR Kit (Cat. No 208052, QIAGEN, Hilden, Germany). The annealing temperature was 60 °C for all customized primers. qPCRs were performed in a Roche LightCycler^®^ 480 II instrument (Roche, Basel, Switzerland). Reactions were at the following conditions: a PCR initial activation step for 15 min at 95 °C; denaturation for 15 s at 94 °C; annealing for 30 s at 60 °C; and extension for 30 s at 72 °C. The acquisition of fluorescence was performed for 50 cycles. PCR efficiencies, a melting curve analysis, and the expression rate were calculated using LightCycler^®^ 480 software (Roche, Basel, Switzerland) as previously described [13]. 

#### 4.6.4. Statistical Analysis

The relative RNA expression and the normalization have been performed as previously described [7,16,51]. Analysis of qRT-PCR data was performed with an unpaired *t*-test considering statistically significant *p*-values < 0.05 (Figure 5). The statistical analysis and related graphs were prepared using GraphPad Prism Software v9.4.0 (GraphPad Software, San Diego, CA, USA). All experiments were conducted at least in biological and technical triplicate.

## Figures and Tables

**Figure 1 antibiotics-13-00264-f001:**
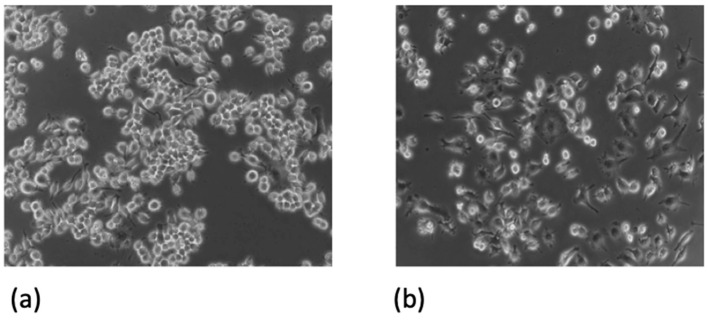
(**a**) Non-infected RAW 264.7 macrophages and (**b**) infected RAW 264.7 macrophages. Microphotograph of RAW 264.7 murine macrophages infected with USA300 ATCC BAA-1556_Catania *S. aureus* at 24 h post-infection. The microphotographs were captured with a Leica DMI 4000B inverted microscope at 20× magnification.

**Figure 2 antibiotics-13-00264-f002:**
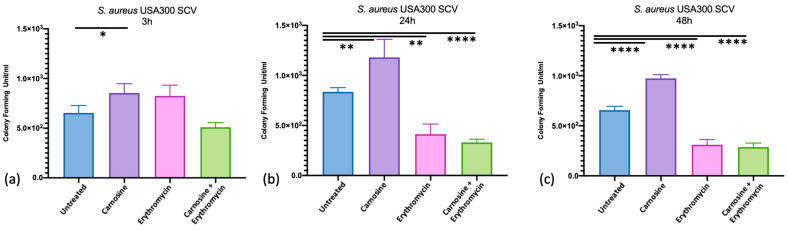
Evaluation of SCVs formation. Small colony variant cells were counted at 3 h (**a**), 24 h (**b**), and 48 h (**c**) post-infection of RAW 264.7 macrophages treated with 20 mM carnosine, 32 mg/L erythromycin, or a combination of both compounds at the same concentration and compared with untreated cells. Bacterial cell counts are reported in histograms as the mean value of colony forming units/mL; bars indicate the statistically significant differences. The number of CFUs were compared with a one-way ANOVA (Nonparametric test) test with corrections for multiple comparisons by controlling the false discovery rate trough the two-stage step-up method of the Benjamini, Krieger, and Yekutieli test, * *p* < 0.05; ** *p* < 0.01; **** *p* < 0.0001.

**Figure 3 antibiotics-13-00264-f003:**
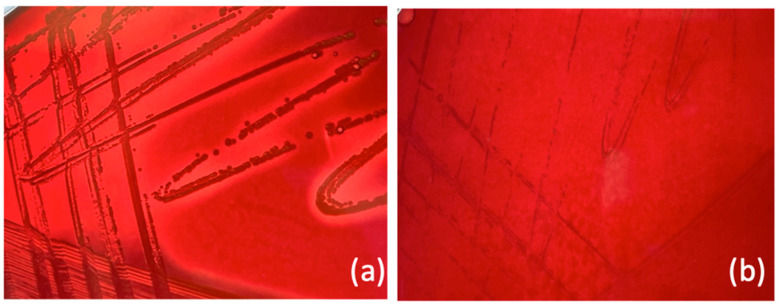
Evaluation of hemolysis on blood agar. (**a**) *S. aureus* wild-type colonies with hemolysis; (**b**) stable SCVs with no evidence of hemolysis.

**Figure 4 antibiotics-13-00264-f004:**
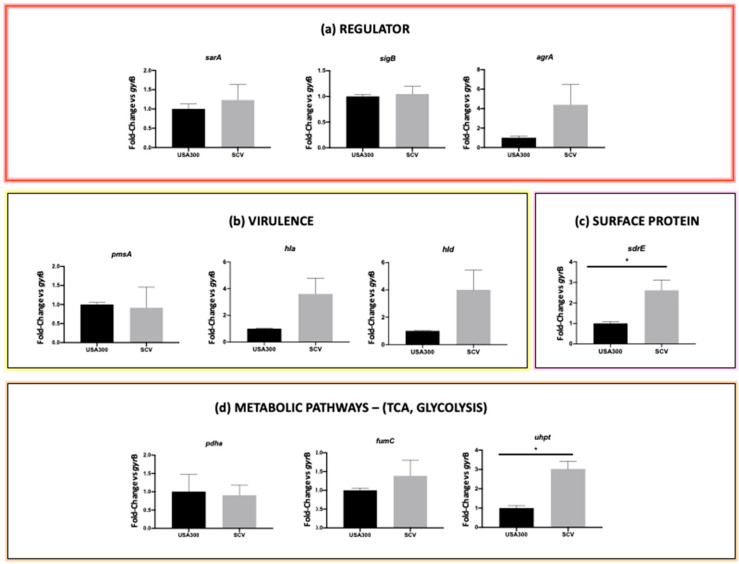
Comparative gene expression analysis. qPCR was used to analyze gene expression in *S. aureus* strain BAA-1556_Catania and its stable SCV derivative isolated after treatment with a combination of 20 mM carnosine and 32 mg/L erythromycin for 48 h. The genes that were analyzed belonged to the following functional categories: (**a**) transcriptional regulators, (**b**) virulence factors, (**c**) surface proteins, and (**d**) glucose metabolism. The *p*-values < 0.05 resulting from the *t*-test were considered statistically significant and are reported in the figure; * *p* < 0.05.

**Figure 5 antibiotics-13-00264-f005:**
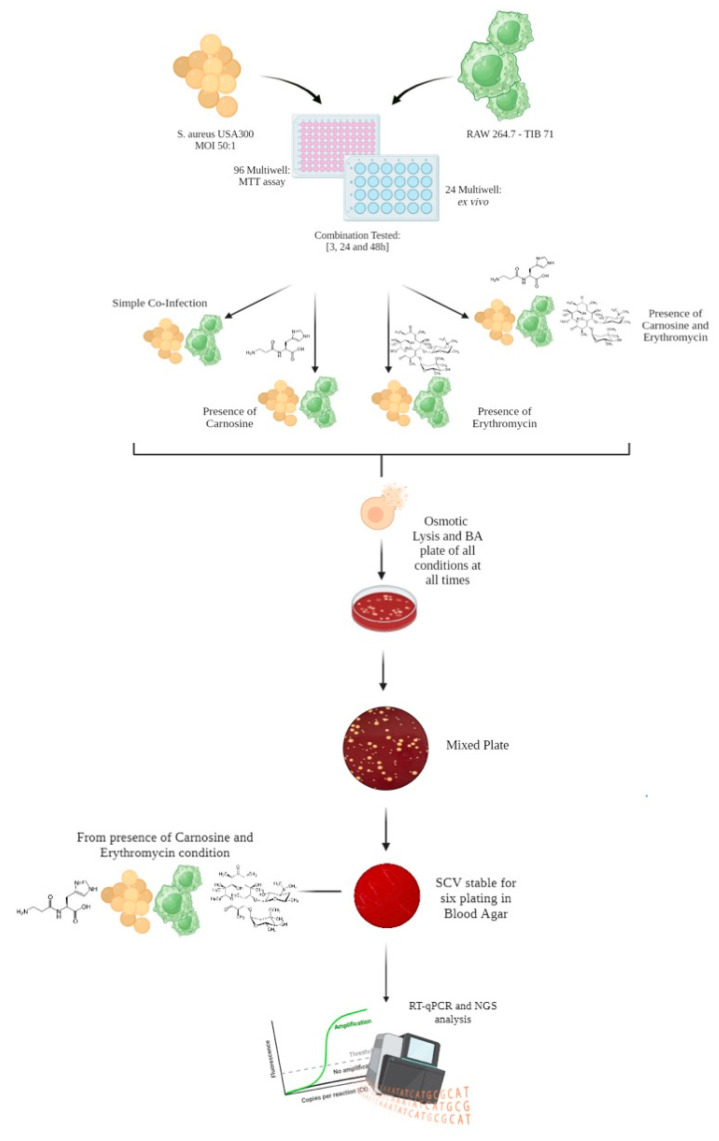
Experimental workflow: from infection to SCV isolation and characterization.

**Table 1 antibiotics-13-00264-t001:** Nucleotide changes obtained by comparing the genomes of *S. aureus* BAA-1556_Catania and its SCV derivative.

Gene	Gene Product	Genome Nucleotide Position	Nucleotide Position in Cds	Codon	Amino acid Change	Type of Mutation
*purR*	Pur operon repressor	532,783	**T**803**A**	A**T**C>A**A**C	**I**267**N**	Missense
Intergenic	-	909,332	C>T	**-**	**-**	**-**
*aroK*	Shikimate kinase	1,659,182	**G**306**A**	T**G**G>T**A**G	**W**102*****	Nonsense

## Data Availability

The datasets presented in this study can be found in online repositories. The names of the repository/repositories and accession number(s) can be found at: https://www.ncbi.nlm.nih.gov/bioproject/?term=PRJNA912391 (accessed on 8 February 2024).

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
