# Peer review of "Generation and Characterization of Stable Small Colony Variants of USA300 Staphylococcus aureus in RAW 264.7 Murine Macrophages"

_antibiotics, 2024, doi:10.3390/antibiotics13030264_

Round 1

Reviewer 1 Report

Comments and Suggestions for Authors

This article contributes to the scientific literature by studying a pathogen of relevant importance for public health. The paper presents a diversified and structured methodology with the results. The discussion is well written and supported with updated references. However, to be accepted for publication, authors must include more details about the importance of carnosine in the introduction. Why did you choose to use carnosine? Have other scientific studies used carnosine in studies with S. aureus or another pathogen? If yes, quote them!

Author Response

REV 1

This article contributes to the scientific literature by studying a pathogen of relevant importance for public health. The paper presents a diversified and structured methodology with the results. The discussion is well written and supported with updated references. However, to be accepted for publication, authors must include more details about the importance of carnosine in the introduction. Why did you choose to use carnosine? Have other scientific studies used carnosine in studies with S. aureus or another pathogen? If yes, quote them!

Thank you for your comment. We implemented the paragraph as suggested (line 71-81) explaining what drives us to use this molecule.

Reviewer 2 Report

Comments and Suggestions for Authors

This manuscript describes the isolation of small-colony variants (SCV) of a S. aureus strain after uptake into mouse macrophages.

The manuscript is generally well presented but provides limited novelty to the field. There are also some questions about the design and execution of the study.

1.        Why were mouse instead of human macrophages used?

2.        There is no description of Fig 1 in the text or figure legend. What do the images show?

3.        Where there only SCV found after plating (figure 2). If not, the number of regular cells should also be shown.

4.        Line 119: please clarify how many passages.

5.        Line 121 and figure 3: why were only the bacteria from the combined treatment tested for hemolytic activity and not the erythromycin only-treated group?

6.        Only one bacterial isolate was selected for WGS (plus control strain). From what treatment group? 

7.        Line 148: what are those two loci encoding for?

8.        Line 164: how where the 10 genes selected? Eg why sdrE and not any other surface protein expressing gene? 

9.        Fig 4: it appears as only one SCV was used for analysis. It would be interesting to see how consistent this expression pattern is over time and passages. Stable vs transient SCV?

Minor points

1.        Table S2; SCVs should be singular (SCV) as only one isolate was sequenced.

2.        Table S3: only 8 of the 10 analysed genes are shown. SdrE is not shown and it looks statistically significant.The top of the table seems to be cut off. What do the numbers represent?

3.        Line 115: what is ‘xxx test’?

4.        Fig 4 legend. Mixed fonts

5.        Line 226: two identical references [23]

6.        Line 446: does this refer to biological or technical triplicates?

Author Response

This manuscript describes the isolation of small-colony variants (SCV) of a S. aureus strain after uptake into mouse macrophages.

The manuscript is generally well presented but provides limited novelty to the field. There are also some questions about the design and execution of the study.

  1. Why were mouse instead of human macrophages used?

Thank you for the comment. The choice of cell line has not been casual. An already published study of Caruso et al reported that the use of carnosine on RAW264.7 murine macrophages protects macrophages cell, modulates the oxidative stress and decreases the expression of genes related to inflammation and pro- and antioxidant systems. DOI: 10.3390/biomedicines9050477 Basing our experiment on this assumption, the treatment of RAW 264.7 murine macrophages with carnosine was finalized to counteract the oxidative stress and support the macrophage response to S. aureus infection.

  1. There is no description of Fig 1 in the text or figure legend. What do the images show?

Thank you for the comment. A brief description of the figure 1 was added in the text (line 102-104).

  1. Where there only SCV found after plating (figure 2). If not, the number of regular cells should also be shown.

Thank you for your comment. We have added a graph reporting comprehensive colony count to the supplementary section, with a cross-reference provided in line 120.

  1. Line 119: please clarify how many passages.

Thanks for the comment. Examining the literature about S. aureus SCVs, we found out that the method to define stable SCVs through cultivation on agar plates is not well established and/or unambiguous. Indeed, different research groups used different methods (3 subculturing passages on TSA, https://www.jstor.org/stable/30126199; 10 subculturing passages on TSA, https://doi.org/10.1128/iai.02702-14; 40 subculturing passages on Blood Agar, https://doi.org/10.1007/s12223-010-0089-3)and so on.
Although the combination of carnosine and erythromycin triggers and supports the mechanisms underlying SCV formation, leading to longer stability compared with other experimental conditions. Our aim is not to assess exactly the duration of the stability but to assess if the stability induced by the combined treatment (erythromycin plus carnosine) would be longer than the one induced by other experimental conditions.

  1. Line 121 and figure 3: why were only the bacteria from the combined treatment tested for hemolytic activity and not the erythromycin only-treated group?

Thanks for the comment. The combined treatment, carnosine plus erythromycin, led to stable SCV isolation. Although reduced hemolysis and increased MIC values for aminoglycosides are features of SCV, our goal was a well characterization of the isolated stable SCV Vs WT strain. SCVs isolated from all other experimental condition showed a transient change in phenotype: the small colonies reverted to the normal phenotype after one passage on blood agar, so we retained that additional test on these strains were uninformative.

  1. Only one bacterial isolate was selected for WGS (plus control strain). From what treatment group?

Thanks for the comment. We selected USA300 ATCC BAA-1556_Catania S. aureus and its derivative stable SCVs from the combined treatment (carnosine plus erythromycin) 48hours post-infection.

  1. Line 148: what are those two loci encoding for?

More information about the products from the two loci are included in the table 1. The CT transition (position 909,332) occurred in an intergenic region between RZS17_04450 and RZS17_04455 instead the GA transition in the coding sequence of aroK (position 1,659,182) caused a nonsense mutation which introduces a premature stop codon.

  1. Line 164: how where the 10 genes selected? Eg why sdrE and not any other surface protein expressing gene?

According to previous studies, we selected genes potentially involved in SCV formation. Once S. aureus USA300 small colony variants (SCVs) were obtained, the main objective of the authors lain in agr locus expression and potential regulation as reported in our study. In general, SCVs have consistently shown reduced agr activity associated with enhanced survival of SCVs within host cells.
Moreover, there is growing evidence that the formation of SCVs could also be due to regulatory mechanisms, involving global regulators (e.g. sigB, sarA and agr), Clp ATPases (Kahl et al, 2005; Mitchell et al, 2008, 2010a) or non-protein-coding RNAs as regulatory molecules (Abu-Qatouseh et al, 2010). [DOI: 10.1002/emmm.201000115 Tuchscherr et al. 2010].

In this work, we tried to select genes linked each other in term of regulation.  For instance, the regulator SarA simultaneously governs the expression of sdrE – involved in the adhesion mechanism – and virulence genes (hla and hld genes), determining the strategy adopted by the bacterium. The same criterion was followed to choose every target included in the study.

  1. Fig 4: it appears as only one SCV was used for analysis. It would be interesting to see how consistent this expression pattern is over time and passages. Stable vs transient SCV?

As already reported, unfortunately a transient phenotype was reported for SCVs isolated from the other experimental conditions except for the combined treatment (carnosine plus erythromycin – 48h p.i.). In order to assess the phenotype stability, passages on blood agar were performed. Following this approach, the transient phenotype was observed only once and, making it difficult to collect for expression studies.

Minor points

  1. Table S2; SCVs should be singular (SCV) as only one isolate was sequenced.

Thanks for the comment. We made the correction.  

  1. Table S3: only 8 of the 10 analysed genes are shown. SdrE is not shown and it looks statistically significant. The top of the table seems to be cut off. What do the numbers represent?

Thanks for the comment. The statistical insight relative to qRT-PCR have been reported in the table. A substitution of table has been done, including all the targets.  

  1. Line 115: what is ‘xxx test’?

Thanks for the comment. The used test has been added (line 130-133)

  1. Fig 4 legend. Mixed fonts

Thanks for the comment. The font has been fixed.

  1. Line 226: two identical references [23]

Thanks for the comment. One out of two references has been deleted.

  1. Line 446: does this refer to biological or technical triplicates?

Thanks for the comment. We refer to both biological and technical replicates. It has also been added in the text (line 463-464).

Reviewer 3 Report

Comments and Suggestions for Authors Bivoni et al have worked on understanding the characteristics of small colony variants of S. aureus. They employed carnosine to aid with the investigation. The authors also looked into whole genome sequence and quantitative gene expression of SCV. While the overall article touches on an interesting subject, there are a few details lacking. Page 2 Line 69 - A more detailed description of carnosine would be required here (or in discussion). Why did the authors only choose carnosine when there are other antioxidants present? Which tissues is it prevalent in? Does it have a specific physiological role?   In other sections - authors talk about SCV. How does SCV related to biofilms? S aureus is known to form biofilms? Do biofilm disruptors affect SCV formation?   The authors look into the stability of SCV by passage. Have they observed a difference between the phenotype and genotype of earlier passages vs later passages? How did they select the final passage?   This paper seems to not really contribute to understanding antibiotic resistance mechanisms. Could the authors maybe add more details in the discussion to compare the gene sequence and expression analysis information and compare it with the currently available antibiotics to add value to the manuscript

Author Response

Bivona et al have worked on understanding the characteristics of small colony variants of S. aureus. They employed carnosine to aid with the investigation. The authors also looked into whole genome sequence and quantitative gene expression of SCV. While the overall article touches on an interesting subject, there are a few details lacking. Page 2 Line 69 - A more detailed description of carnosine would be required here (or in discussion).  Why did the authors only choose carnosine when there are other antioxidants present? Which tissues is it prevalent in? Does it have a specific physiological role? 

Thank you for your comment. A more detailed description of carnosine has been added in the introduction section as suggested (line 71-81). An already published study of Caruso (one of the co-author) et al reported that the use of carnosine on RAW264.7 murine macrophages protects macrophages cell, modulates the oxidative stress and decreases the expression of genes related to inflammation and pro- and antioxidant systems. DOI: 10.3390/biomedicines9050477 Basing our experiment on this assumption, the treatment of RAW 264.7 murine macrophages with carnosine was finalized to counteract the oxidative stress and support the macrophage response to S. aureus infection. Furthermore, being carnosine naturally occurring in high concentration in mammalian skeletal muscles, no toxic effect has been showed when added to RAW 264.7 murine macrophages.

In other sections - authors talk about SCV. How does SCV related to biofilms? S aureus is known to form biofilms? Do biofilm disruptors affect SCV formation?   The authors look into the stability of SCV by passage. Have they observed a difference between the phenotype and genotype of earlier passages vs later passages? How did they select the final passage?   This paper seems to not really contribute to understanding antibiotic resistance mechanisms. Could the authors maybe add more details in the discussion to compare the gene sequence and expression analysis information and compare it with the currently available antibiotics to add value to the manuscript.

SCV formation and biofilm generation are two different aspects of the adaptive life of S. aureus. In both cases, and for different reasons, these alternative lifestyle forms result in bacteria non-susceptible to the treatment with antibiotics, giving a resistant phenotype. Furthermore, the condition to induce biofilm formation in S. aureus require adhesion to biotic or abiotic surfaces and a profound regulation activated when a specific bacterial concentration is reached.

SCVs are cells with reduced colony size that arise from the wild-type strain in conditions associated with chronic infections. The acquisition of the SCV phenotype could be determined by changes at transcription level - unstable SCV phenotype - or at genetic level – stable SCV phenotype depending on the environmental conditions. Despite the acquired features of the SCV phenotype over time, the presence of SCVs poses a challenge in treating infection: SCVs do not respond to the antibiotic therapies, becoming an infectious reservoir.

Round 2

Reviewer 2 Report

Comments and Suggestions for Authors

The authors have addressed the points I raised and provided satisfactory explanations. However, these need to be explained to the readers as well to clarify the limitations of this study (points 1, 4-9).

  1. There is no description of Fig 1 in the text or figure legend. What do the images show?

Thank you for the comment. A brief description of the figure 1 was added in the text (line 102-104).

This is just a repeat from what is written in the figure legend. What are the differences in the two phenotypes and how can they be explained?

Author Response

The authors have addressed the points I raised and provided satisfactory explanations. However, these need to be explained to the readers as well to clarify the limitations of this study (points 1, 4-9).

An implementation of the aforementioned three points was provided (line 108-113; line 144-148; line 185-198).

  1. There is no description of Fig 1 in the text or figure legend. What do the images show?

Thank you for the comment. A brief description of the figure 1 was added in the text (line 102-104).

This is just a repeat from what is written in the figure legend. What are the differences in the two phenotypes and how can they be explained?

Thank you for the comment. An explanation was reported (line 104-106).